# Crossmodal interference on counting performance: Evidence for shared attentional resources

**Claudia Del Gatto** [1]*, **Allegra Indraccolo**[1], **Tiziana Pedale**[2,3], **Riccardo Brunetti**[1]

**1** Experimental and Applied Psychology Laboratory, Department of Human Sciences, Università Europea di Roma, Rome, Italy, **2** Department of Physiology and Pharmacology, Sapienza University of Rome, Rome, Italy, **3** Functional Neuroimaging Laboratory, Fondazione Santa Lucia, IRCCS, Rome, Italy

* claudia.delgatto@unier.it

**Data Availability Statement:** All relevant data are within the paper and its Supporting Information files.

## Abstract

During the act of counting, our perceptual system may rely on information coming from different sensory channels. However, when the information coming from different sources is discordant, such as in the case of a de-synchronization between visual stimuli to be counted and irrelevant auditory stimuli, the performance in a sequential counting task might deteriorate. Such deterioration may originate from two different mechanisms, both linked to exogenous attention attracted by auditory stimuli. Indeed, exogenous auditory triggers may infiltrate our internal "counter", interfering with the counting process, resulting in an overcount; alternatively, the exogenous auditory triggers may disrupt the internal "counter" by deviating participants' attention from the visual stimuli, resulting in an undercount. We tested these hypotheses by asking participants to count visual discs sequentially appearing on the screen while listening to task-irrelevant sounds, in systematically varied conditions: visual stimuli could be synchronized or de-synchronized with sounds; they could feature regular or irregular pacing; and their speed presentation could be fast (approx. 3/sec), moderate (approx. 2/sec), or slow (approx. 1.5/sec). Our results support the second hypothesis since participants tend to undercount visual stimuli in all harder conditions (de-synchronized, irregular, fast sequences). We discuss these results in detail, adding novel elements to the study of crossmodal interference.

## Introduction

Event counting, such as figuring out how many drops of medicine we are putting in a glass of water or how many times cards of a specific suit appeared during a card game, is a common everyday activity used in many different situations. Logie and Baddeley [1] suggested that event counting engages three types of cognitive processes: firstly, we perceive the item to be counted; then, we retrieve a proper counting system from long-term memory; and, finally, the working memory support the process of maintaining the running total. More specifically, they proposed the phonological loop to be the working memory sub-system responsible for both

**Funding:** The author(s) received no specific funding for this work.

**Competing interests:** The authors have declared that no competing interests exist.

the process of subvocal articulation of numbers and the maintenance of the running total [2]. However, the process of counting can also rely specifically on attentional resources, especially when it comes to selective counting (e.g., telling apart target items to be counted from distractors; [3, 4]).

The first cognitive process involved in the process of counting, as stated by Logie and Baddeley [1] is the perception of the event or stimulus to be counted. There are many situations where we are simultaneously presented with information from multiple sensory modalities and crossmodal congruence or incongruence has a great impact on perception [5]. Many studies have demonstrated the existence of crossmodal interplay between auditory features and visual dimensions, such as spatial elevation [6–11], brightness [12, 13] or lightness [13–17], size [8, 17, 18], angularity of shape [13], direction of movement [19], synchrony, and temporal congruence [20].

Although a wide range of research has focused on crossmodal interactions across all five senses, at a perceptual, mnemonic, or motor level (see for example [21–25]), to the best of our knowledge, no studies have directly investigated crossmodal interactions in event counting.

As mentioned, the second system involved in the counting processes is memory [1]. In this case as well, the impact of crossmodality seems to be relevant: for example, it has been shown that irrelevant background speech disrupts serial short-term memory for materials presented visually [26]. Other studies have shown the impact of crossmodal effects on working memory performances. For example, a study by Brunetti et al. [27] used an audiovisual two-back task with different kinds of audiovisual correspondences in congruent and incongruent conditions, showing better memory performances in congruent conditions. Interestingly, it seems that when stimuli from different sensory modalities offer additional information, they can facilitate processing [28–32]. However, there are also situations where stimuli presented simultaneously in different modalities can impair performance: When sensory modalities offer conflicting information, stimuli presented in one modality can alter or attenuate processing in the other modality [33–37].

## Sensory dominance

Much research of this kind, directed for the most part in the domain of attention, pointed out visual dominance. For instance, in the Colavita visual dominance task [33], participants were asked to press a button in response to a sound and another one in response to a light. Interestingly, participants often press only the visual button when both stimuli are presented together, suggesting that the visual modality dominates when responding to multisensory information [30, 38] (see [39] for a review). On the other hand, further research shows that when temporal judgment is required auditory processing can dominate over visual processing. Shams and colleagues [35] and also Burr and Bank [40] showed that the perception of visual stimuli can be distorted by the simultaneous presentation of sounds. In a classic experiment examining the flash illusion, participants see flashes of light while also hearing sounds presented in fast temporal sequences. When participants are presented with a single flash and two sounds, they tend to perceive more than one flash (fission illusion). Interestingly, although weaker, the opposite phenomenon was also observed, i.e., the so-called fusion illusion, where the combination of two visual stimuli and a single auditory stimulus might induce the perception of a single visual stimulus (see, for reviews, [41, 42]). In the first case, the final effect is an overcount of the visual stimuli, while in the second scenario the visual stimuli tend to be undercounted. Studies focusing on the neural mechanisms underlying the fission illusions (EEG, ERP, and fMRI studies) evidenced that this phenomenon is associated with increased gamma band power [43], increased evoked responses [44, 45], and increased bold activity in the primary

visual cortex, and in the superior temporal sulcus, a key region involved in multisensory integration [46, 47]. As for the primary visual cortex, it is interesting that when irrelevant auditory stimuli effectively produced a fission illusion, suggesting an "overcount mechanism", the occipital visual cortex appeared to increase its activity [46, 47]. On the contrary, during the fusion illusion, suggesting an "undercount" effect, a reduced BOLD signal in the same occipital visual regions was observed [47]. All these findings support an auditory dominance during crossmodal processing, which appear to dramatically impact on the visual perception at behavioral and neural level. The same asymmetry was found in the context of the counting process. Abadi and Murphy [48], studied both fission and fusion illusions by examining counting performance with stimuli that could be spatially and temporally variable. They found sound-induced fusion to be more robust than sound induced fission and that sound-induced fusion is the most likely outcome when visual and auditory stimuli are incongruent (e.g. presented at different time intervals). In this case, therefore, the visual dominance is variable. These results have been interpreted by the authors with the concept of 'auditory driving', in which task-irrelevant auditory stimuli can influence the number of visual targets either by excess or by defect [49–52]. In line with this, Ljung and colleagues [53] also investigated the effects of auditory stimuli in a counting task. Specifically, the authors examined if the meaning of a background speech could affect event counting. They suggested that when background speech is similar in meaning to the target task process, it contributes to the disruption. The topic of auditory dominance is further debated, indeed, in a study by Ngo, Cadieux et al. [54]: while some of their results highlight auditory dominance, other ones reveal visual dominance. In the field of sensorimotor literature, research suggests that rhythmic motor behavior is indeed attracted more strongly to auditory than to visual sequences. When there is interference between auditory and visual stimuli, participants' evaluations are usually more biased toward the auditory than the visual temporal information [35, 55–64].

A study by Dunifon et al. [65] reported a more complex pattern of modality dominance, finding evidence for both auditory and visual dominance. Results showed that crossmodal presentation deteriorated auditory discrimination but also visual reaction times: The authors interpret these outcomes suggesting that auditory stimuli automatically engage attention and attenuate visual processing.

## Mechanisms of crossmodal interference

When stimuli are presented bimodally, we can find many examples of crossmodal interference: e.g., different task-irrelevant auditory stimuli seem to have detrimental effects on the immediate recall of visually presented materials, by interfering with working memory processes [26, 66–71]. In a study by Sinnett and Kingstone [72], participants had to evaluate the direction of a tennis ball thrown in two different conditions, without any sounds or paired with a grunting noise. Results showed that the grunting sound condition slowed reaction time and increased errors rates compared with the silent condition. The detrimental effect of task-irrelevant auditory stimuli was also corroborated by Robinson and Parker [73]: They showed that response times to visual stimuli were slower when the visual sequences were presented together with tones. Also, Laughery, Pesina and Robinson [74] examined response times and eye tracking variables while participants completed variations of unimodal and crossmodal serial response time tasks. They found that participants were slower to fixate on the visual stimuli in the crossmodal condition compared to the unimodal condition.

Another study found crossmodal interference particularly in the process of counting. Lo and Lay [75] used a counting task with the presence of either congruent or incongruent sensory signals in the background. Participants were asked to count the number of flashes of a

black circle or the presentation frequency of a sound. Conjointly background sensory cues were presented which could be congruent or incongruent with respect to the targets. Results demonstrated that participants generally performed better with congruent than with the incongruent stimuli.

On the other hand, other studies go in the opposite direction and demonstrate increased visual processing inducted by sounds [76, 77]. For example, McDonald et al. [78] provides psychophysical evidence that a task-irrelevant sound facilitates the detectability of a following flash appearing at the same position. Also, Stein et al. [79] asked participants to rate the intensity of a visual stimuli that could be accompanied by an auditory stimulus or presented alone. Results showed that the auditory stimulus improved the evaluation of visual stimuli.

In the cases seen so far, auditory stimuli appear to interact in complex ways with visual stimuli. In some cases, it seems that visual dominance emerges [33], in others instead it emerges auditory dominance [35, 40], in other cases a more complex pattern of dominance come up [65] and again, in the sensorimotor field it seems that when movement or its planning is involved, auditory dominates over visual stimuli [60–63]. These complex patterns of results are also dependent by the way in which different modalities may capture our attention.

## Attention in multisensory processing

It is important to underline the role of attention in the counting process as well as its role in the processing of multisensory stimulation. Mainly, attention facilitates working memory as a buffer for both the encoding and retrieval of information necessary for performing cognitive tasks like counting [3]. Selective attention is in charge of managing new information during various phases of processing, moreover, can act as a supervisor for working memory by enhancing the encoding of the information more relevant to the specific task at hand.

Regarding the role of attention on multisensory stimuli, a review [80] highlighted the effect of top-down and bottom-up processes. Indeed, some studies have reported that multisensory bottom-up processes have the capability to capture our attention [81]. On the other hand, other studies [31, 82] focused on how other higher-level factors can impact on integration of information across the senses. Talsma et al. [80], propose that multisensory integration tends to be pre-attentive when the amount of competition between resources is low. In contrast, when stimuli between different modalities are competing for resources, and thus the saliency is low, top-down selective attention could be necessary [83, 84]. A corollary of this explanation is that auditory and visual modalities share the same pool of attentional resources and compete for them: a supramodal attention that can be allocated to whatever sensory modality in bottom-up or top-down way (see [85–87]).

The supramodal approach relies on executive functions: The system dedicated to managing conflicting multisensory messages coming from irrelevant and relevant sources [88–92]. This system has been shown to be supramodal by integrating stimuli coming from different modalities [91–94].

In contrast to a supramodal nature point of view, each sensory modality could have its own separate pool of attentional resources (the modal view: [95]). Therefore, separate tasks performed in two different sensory modalities would not interfere. Evidence from behavioral studies support this assumption [96]. Keitel et al. [97], for example, investigated the consequences of a shift from simultaneous bimodal (visual and auditory) to unimodal selective attention. Their results suggest a modality-specific resources account. Given the conflicting experimental evidence, the debate on modality specific or supramodal nature of attentional resources is still open. However, in the specific context of the counting process it is possible to hypothesize that a type of supramodal attention intervenes since it has already been

demonstrated that there are crossmodal interferences in tasks involving counting: for example when visual target stimuli are presented simultaneously with incongruent auditory stimuli [75].

## Factors affecting selective counting

The present study investigates how the visual and auditory modalities interact in a longer, continuous task involving attentional and ideomotor simulations aspects. Specifically, a crossmodal counting task, although managed by working memory, undergoes attentional interference. Indeed, crossmodal interactions between visual and auditory stimuli are possibly still effective in a continuous process, such as counting. Verbal counting is a well-learned ability; however, it can be difficult and error-prone, especially for large magnitudes: with an increasing number of objects to be counted it is easy to lose track of the running total.

An interesting result with a counting task in the crossmodal domain has been obtained by Carlyon and colleagues [98]. In this investigation, participants heard a sequence of tones and saw a sequence of visual stimuli; their task was to count, according to the request, either the number of visual or the number of auditory stimuli. The results showed a better performance when participants counted the auditory targets than in all the other conditions, confirming that the auditory dominance found in other temporal tasks applies to counting as well.

The counting process seems to be affected also by unimodal variables, for example, the so-called fluency, which is the timing regularity of the events to be counted. Carlson and Cassenti [99] examined specifically these effects: they asked participants to count events occurring at regular or irregular intervals and then to report their total count. In general, results showed that participants counted regular events more accurately. Stevenson and Carlson [100] investigated fluency as well: They asked participants to count asterisks as they appeared on the screen with regular (isochronous) or irregular Stimulus Onset Asynchronies (SOAs). Then, participants were asked to provide a final count and a confidence rating. Their results showed that participants were more accurate and produced greater confidence ratings with regular timings.

The superiority of regular sequences can be explained by the fact that temporal regularity between events can also greatly improve our perception as highlighted by the extensive work of Kia Nobre and Jennifer Coull [101]. Moreover, the Dynamic Attending Theory (DAT; [102–104]) suggests that attention is not allocated regularly in time but fluctuates periodically according to internal dynamic oscillators. These oscillators regulate an individual's attending rhythm and thus the creation of expectation over time. Accordinlgy, a study by Escoffier et al. [105], compared behavioral responses to visual stimuli presented in and out of synchrony with a musical rhythm and showed faster reaction time for picture presented with the musical rhythm and on-beat. Furthermore, it has been shown that crossmodal entrainment in rhythmic multisensory stimuli can impact behavioral outcomes of perception. In a recent study Lin, Yeh, and Shams [106] investigated the effect of congruence between visual and auditory stimuli on perceptual pleasure (positive emotional response to sensory stimuli), in the sense of the rhythmic temporal synchronization between auditory and visual stimuli. The authors found higher perceptual pleasure for temporal congruency compared with incongruency. These results indicate that rhythm entrains attention, even at a subliminal level [106] facilitating information processing in synchrony with the rhythm.

Lastly, another variable that may potentially affect counting is speed. The counting process may be considered as an ideomotor behavior. The ideomotor approach [107, 108], is influenced by the idea of a strong link between perception and action [109] (for crossmodal effects see also, [23, 110]). The idea is that the representation of actions and representation of stimuli

share the same format. However, some studies proved that the ideomotor concept could be much more than a motor mechanism. Indeed, Badets et al. ([111], Experiment 1) highlighted the role of an ideomotor mechanism in number processing without any movement [112]. Thanks to these studies it is possible to hypothesize that the activity of counting silently (covert counting), can be connected to an ideomotor simulation in which a process of subvocal numerical representation is activated, thanks to the articulatory process of the phonological loop [99]. In other words, even if it is not required to count aloud, counting may have limitations related to the internal articulation of numbers. If this is the case, we would expect to find significant effects connected to speed, as the articulatory process, even subvocally, is constrained to articulation limits (see for example [113, 114], regarding articulatory limitations related to the time required to pronounce some words in different languages). Moreover, considering the ideomotor account with respect to subvocalic repetition during a counting process, it is possible to hypothesize that when the items to be counted appear very quickly and their magnitude increases, performance may deteriorate, also because participants has not yet finished subvocalizing the current count that already appears the following stimulus to be counted.

Finally, counting processes can be affected by our preferred tempo. Indeed, "Spontaneous motor tempo", namely the tempo of self-paced regular and repeated movements, that corresponds to the preferred and natural rhythm to execute isochronous motor actions (e.g., walking, or tapping a finger), which usually tends to cluster around 500–600 msec in adults [115–117] might modulate our performance in counting. To the extent of our knowledge, to date no one has explicitly investigated these aspects in counting, thus we find it important to include the variable of speed to study its effects in association with other variables that can influence the counting process. For example, the interplay between speed and fluency could yield interesting results. Since regularity is an important factor in a counting task [99, 100] it would be interesting if this could counterbalance the difficulty that might come with speed.

Counting can thus be a task prone to errors due to different crossmodal or unimodal features [74] as well as temporal variables such as regularity [99]. In this study, we focus on the crossmodal effect of simultaneously presented visual and auditory information on covert event counting (that is—to count internally without verbalizing the count). Our interest is to verify if task-irrelevant auditory information can attract exogenously our attention and disrupt the counting process. This is in line with the idea that crossmodal signals can affect sensory processing by directing attention [35].

In general, we also expect that the variables Speed (slow or fast presentation of stimuli to be counted) and Isochronicity (SOAs regularity or irregularity, or rhythmicity, see Shams et al. [35]) can affect participants' performance. Specifically, the faster the visual events are presented, the more likely it will be to have errors in counting. Based on previous literature, slow presentation implies events (visual and/or auditory stimuli) that happen up to 95 bpm (e.g. 1.5 events every second); moderate presentation implies events that happen from 95 to 150 bpm (in line with the "Spontaneous motor tempo"; e.g. 2 events every second); instead, fast presentation implies events that happen above 150 bpm (e.g. 3 events every second [118, 119]).

In addition, SOAs regularity (isochronicity) should result in a better performance compared to irregular SOAs [99]. Finally, we expect better performance when visual and auditory stimuli are presented synchronously compared to when the presentation is unsynchronized [120].

To summarize, we have considered three different variables that could affect counting processes: 1. Speed—we hypothesize that increasing presentation speed will result generally in worse performance; 2. Isochronicity (regular or irregular time intervals between target stimuli) —we hypothesize that counting sequences featuring regular timing will be easier than counting events with irregular intervals [90, 91] 3. Synchrony—we hypothesize that the presence of

sounds in synchrony with visual stimuli to be counted could facilitate the performance; on the other hand, a condition where sounds are asynchronous with visual stimuli could result in an overcount or an undercount (e.g. [5]).

More specifically, regarding this last variable, we want to test two different hypotheses regarding the possible disruptive effects of Synchrony or the lack thereof. These two hypotheses are linked to the ability of sound to exogenously capture our attention, even if task-irrelevant [121–123]. The first hypothesis is that covert counting of visual stimuli can be distorted by the simultaneous presentation of asynchronous sounds that, by exogenously attracting our attention, can trigger the counting process and infiltrate the current count: This infiltration, where irrelevant asynchronous sounds are able to trigger our counting process more than necessary, would result in an overcount. We hypothesize that this could happen even if the amount of auditory and visual stimuli is the same since the variation in numerosity is given illusorily by asynchrony, in other words the fact that the stimuli are not perceptually aligned could elicit the perception of a different numerosity, just as in Shams' illusion [35]. The second hypothesis, on the other hand, postulates that the task-irrelevant sounds presented asynchronously with the visual stimuli will disrupt the counting process (e.g., [53, 124]): This disruption, by temporarily distracting the counting process, would result in missing some of the visual events and eventually in an undercount. Hence, the results can help us to glean a more complete picture on how asynchronous sounds can affect a visual counting task: if the participants tend to overcount, we can assume the first hypothesis (infiltration) to be true, while if they undercount, the second one (disruption) will be confirmed. To test these hypotheses, we conducted two experiments: We will now illustrate them and then interpret the results of both in the General Discussion.

## Experiment 1

### Method

All methods were carried out in accordance with declaration of Helsinki. The experimental protocol has been approved by the Ethics Review Board of the Università Europea di Roma.

**Participants.** Seventy-nine participants (65 females, mean age = 20.71 years; SD = 1.49 years; range = 19–25) took part in the experiment for course credit. The administrations were carried out between 20 January and 28 February 2022. All the participants reported normal or corrected-to-normal vision and audition. The participants were naïve as to the purpose of the study and gave informed written consent. Authors had access to information (Name and Surname, university registration number, gender, hours of sleep in the night before doing the experiment, age) that could identify individual participants during and after data collection.

**Materials.** Thirty-six video or audio-video sequences were created for the experiment. All sequences were generated at 12 fps (each frame lasted for approx. 84 msec) and featured a flashing white disc (visual angle: approx. 9˚ on a grey (50%) background; the measurement of the visual angle must be considered an approximation because the experiment was administered online, therefore allowing for different screen sizes and distances from the screen). Since the experiment was administered online, even though participant was instructed to sit approximately 60 cm from the screen it was not possible to control this variable accurately. However, the size of the visual stimulus was kept constant during the experiment, and it was irrelevant to the task. Even if the participants varied their distance from the screen to perform the task, we can assume this variation was random, thus not systematically affecting the results.

Each disc flash lasted for 2 frames (168 msec) in all sequences. The number of flashes in each sequence varied pseudo-randomly (average: 31, min: 28, max: 34). In half of the sequences, the flashes were isochronous (regular time interval between stimuli), while in the

other half they were irregular. Half of the sequences were "Moderate" (2 flashes per second on average) while the other half was "Fast" (3 flashes per second on average) and varied in length according to the total number of flashes (average length, in seconds, of "Moderate" sequences: 15.3, min: 13.6, max: 17.4; "Fast" sequences, average: 10.6, min: 9.2, max: 12.8). Sequences varied also according to sound presence and type: "No sound" sequences were only videos with no sounds; "Synchronous" sequences had sounds synchronized with the onset of disc flashes; "Asynchronous" sequences had the same amount of sound and flashes, but they were randomly scattered in the sequence, without any synchronization (i.e., no onset of the discs had a concurring sound onset). The sound used to create the sequences was a clearly audible "tick" lasting for 30 msec. The systematic crossing of these variables resulted in 12 different conditions (see Fig 1): 1. White disc flashing isochronously with no sound, moderate (2/s); 2. White disc flashing isochronously with no sound, fast (3/s); 3. White disc flashing isochronously with synchronous "tick" sounds, moderate (2/s); 4. White disc flashing isochronously with synchronous "tick" sounds, fast (3/s); 5. White disc flashing isochronously with asynchronous "tick" sounds, moderate (2/s); 6. White disc flashing isochronously with asynchronous "tick" sounds, fast (3/s); 7. White disc flashing non-isochronously with no sound, moderate (2/s); 8. White disc flashing non-isochronously with no sound, fast (3/s); 9. White disc flashing non-isochronously with synchronous "tick" sounds, moderate (2/s); 10. White disc flashing

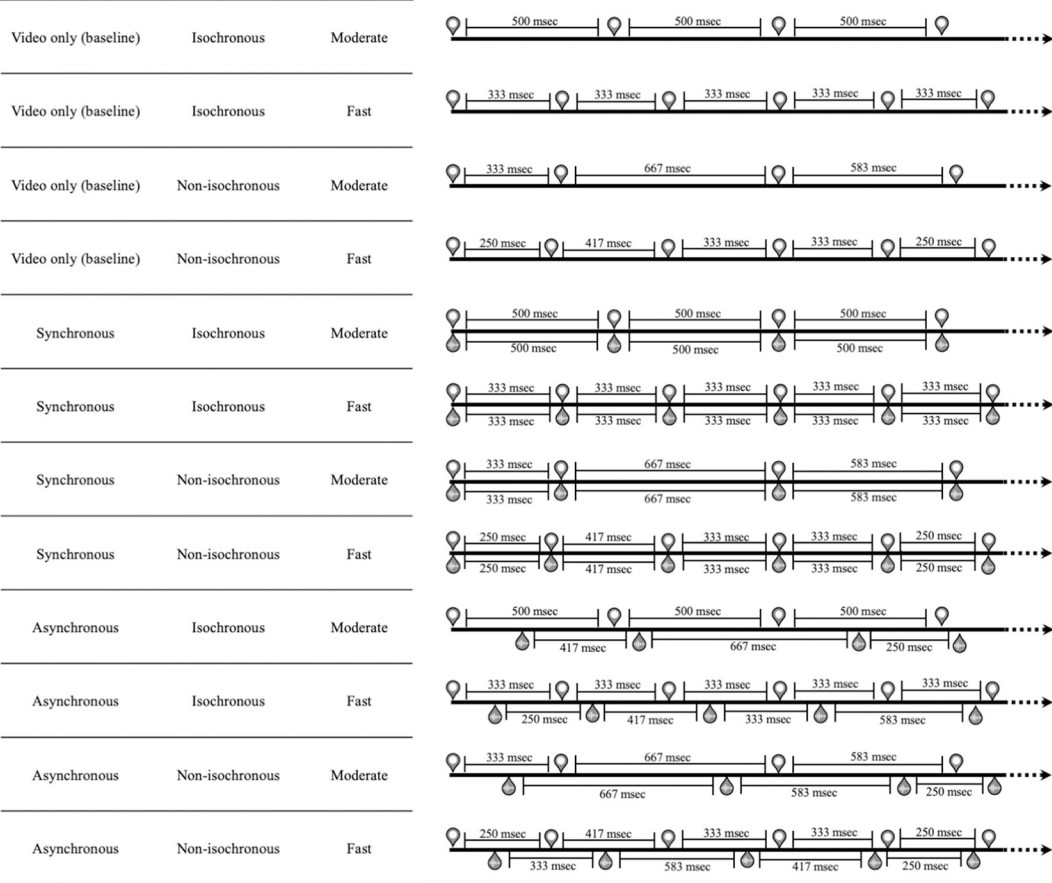

**Fig 1. Examples of sequence types used in Experiment 1.** All icons above the timelines represent visual stimuli (white discs on grey background), while all icons below the timelines represent "tick" sounds. The timings of all stimuli fit in the frames of the 12 fps structure (each frame is approximately 83 msec) used to create the video sequences.

non-isochronously with synchronous "tick" sounds, fast (3/s); 11. White disc flashing non-iso-chronously with asynchronous "tick " sounds, moderate (2/s); 12. White disc flashing non-iso-chronously with asynchronous "tick " sounds, fast (3/s). In the bimodal conditions, the number of visual and auditory stimuli was equal. Each condition had 3 instances, for a total of 36 trials. The 36 trials containing the 12 different conditions were presented in a random order. Every sequence was followed, after a random interval of approx. 2 sec, by 1 out of 4 "control" sounds: gunshot, dog barking, cat meowing, glass shattering. The control sounds had a length varying from 250 msec to 600 msec. The volume of the control sounds was adjusted according to the volume of the auditory stimuli: specifically it was adjusted to a lower volume. In other words, the control sound was always at a lower volume in order to make it audible but to avoid increasing arousal levels.

**Procedure.** Each participant was presented with all the sequences. All sequences ended, a few seconds after the last flashing disc disappeared, with a control sound. The participants were invited to count the flashing white discs that appear in the sequences and to indicate, at the end of the presentation, their count, by clicking on an on-screen scale showing numbers ranging from 15 to 40 and to indicate what kind of sound they heard at the end. The task was the same throughout the whole experiment, which lasted approximately 20 min. After receiving instructions, the participants performed 5 training trials to get used to the task. This training involved both bimodal and unimodal (visual only) stimuli.

Since the experiment was administered online, several measures were taken to make sure they performed the experiment correctly: 1) The recognition of the control sound was included to make sure they had the volume turned on and at a hearable level while performing the task; 2) The interpolation of sequences without sounds was used both to record a unimodal baseline, but also to make sure that the participants would not risk to count sounds instead of flashing discs. No feedback about the correctness of the response was given.

**Apparatus.** Stimulus presentation, conditions, randomization, and the recording of responses were all controlled by a custom-made script in the PsychoPy (v 1.80; [125]) programming environment, with online administration using Pavlovia.org [125].

## Results

To perform the analyses only those participants that clearly heard the sound for the entire experiment we included only the participants with accuracy at the control question above 90%: this led to the exclusion of two participants. Thus, the analysis was performed on 77 participants (10 males; mean age = 20.72 years; SD = 1.51; range 18–25 years). The average accuracy to the control question on this subsample of subjects was 99% (SD = 0.02%) indicating that the participants clearly heard the sounds for the entire duration of the experiment.

The next analyses were performed on Delta values, which were calculated by subtracting the correct count (CC) from the count given by the participant (PC; PC-CC = Delta value). In this way, we obtained a value that indicates the deviation of each participant's count from the correct count: If the value is negative, it means that the participant underestimated the number of stimuli; if the value is positive, it means that the participant overestimated the count. We ran a three-way repeated-measure Bayesian analyses of variance (ANOVAs; [126]) with Iso-chrony (2 levels), Speed (2 levels) and Synchrony (3 levels) as independent within-subject factors, using Delta data as a dependent variable. The advantage of the Bayesian approach is that, while classical frequentist ANOVA allows only to accept or reject the presence of an effect, Bayesian ANOVA permits to recognize the best model describing the dependent variable by means of model comparisons ($BF_M$) and allows to estimate the strength of evidence in favor ($BF_{10}$) of each factor involved in the model according to the following interpretation: (BF) < 1:

**Table 1. Analysis of the effects in the Bayesian ANOVA conducted on the Delta data in Experiment 1.**

| Effects | P(incl) | P(excl) | P(incl\|data) | P(excl\|data) | BF$_{10}$ |
|---|---|---|---|---|---|
| Isochrony | 0.263 | 0.263 | 0.005 | 0.057 | 0.097 |
| Synchrony | 0.263 | 0.263 | 0.706 | $3.24 \times 10^{14}$ | $2.18 \times 10^{13}$ |
| Speed | 0.263 | 0.263 | 0.052 | $2.53 \times 10^{37}$ | $2.041 \times 10^{35}$ |
| Isochrony ✱ Synchrony | 0.263 | 0.263 | 0.117 | 0.803 | 0.146 |
| Isochrony ✱ Speed | 0.263 | 0.263 | 0.914 | 0.006 | 149.184 |
| Synchrony ✱ Speed | 0.263 | 0.263 | 0.171 | 0.805 | 0.213 |
| Isochrony ✱ Synchrony ✱ Speed | 0.053 | 0.053 | 0.023 | 0.017 | 1.328 |

P(incl) and P(excl) represent the priors of including and excluding the predictor before seeing the model. P(incl|data) and P(excl|data) represent the posteriors of including and excluding the predictor after seeing the model. BF$_{10}$ represents how much more likely the data are under the model including the predictor than under the model excluding the predictor.

no evidence, 1–3: anecdotal evidence, 3–10: substantial evidence, 10–30: strong evidence, 30–100: very strong evidence, > 100: decisive evidence [127, 128]. The strength of evidence supporting the inclusion of each factor was evaluated by a comparison between the model that contains the effect of interest with the matched model deprived of the effect. For post hoc tests, correction for multiple comparisons was performed using the approach discussed in Westfall [129], which consists of the following steps (see also [130]): As a first step, Bayesian t-tests are computed for all pairwise comparisons, thus providing unadjusted Bayes factors; as a second step, the prior model odds are adjusted by fixing to 0.5 the prior probability that the null hypothesis holds across all comparisons; finally, the adjusted prior odds and the Bayes factor are used to calculate the adjusted posterior odds. All the statistical analyses were performed using JASP (Version 0.16.4, https://jasp-stats.org/).

Bayesian ANOVA revealed as best model, explaining the Delta data, the model containing the effect of Speed, Synchrony, and Isochrony, and the interaction between Isochrony and Speed (BF$_M$ = 34.12, see also Table 1 and Fig 2 with all the experimental conditions). The evidence in favor of the inclusion of the main effect of Speed and Synchrony was decisive (all BF$_{10}$ > 100). On the contrary, there was no evidence for the main effect of Isochrony (BF$_{10}$ =

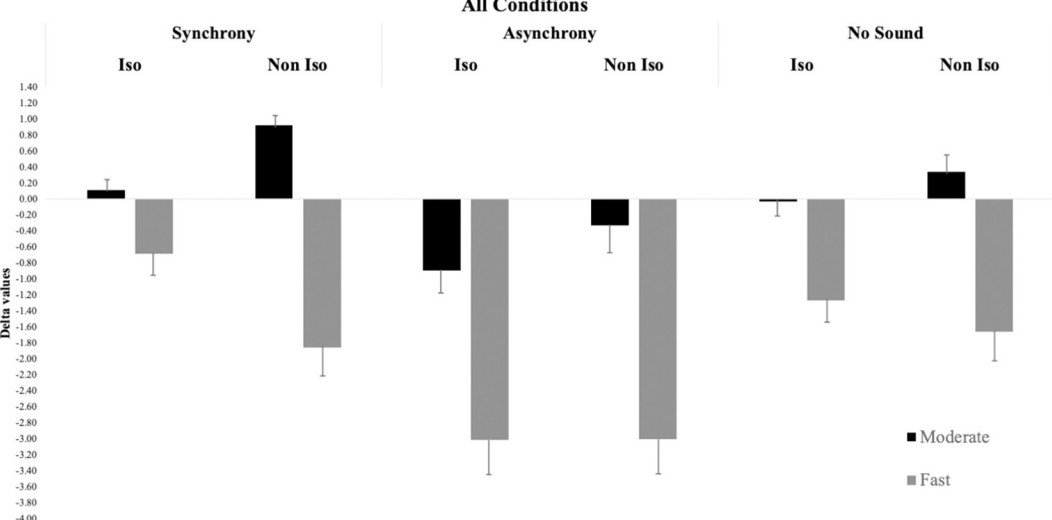

**Fig 2. Bar graph displaying Delta data of all conditions of Experiment 1.** The error bars represent the standard errors.

0.10), but there was decisive evidence in favor of the inclusion of the interaction between Isochrony and Speed ($BF_{10} > 100$). As expected, there was no evidence in favor of the inclusion of all the other interactions ("Speed*Synchrony", $BF_{10} = 0.21$; "Isochrony*Synchrony", $BF_{10} = 0.15$; and the three-way interaction "Speed*Synchrony*Isochrony" $BF_{10} = 1.33$).

Post-hoc analysis of the Speed effect revealed that participants were more accurate when the presentation of stimuli was at a moderate speed (Delta value: -0.08) as compared to the faster condition (Delta value: -2.11), ($BF_{10} > 100$, posterior odds = $1.79 \times 10^{34}$). When the presentation of stimuli was faster participants undercounted the stimuli, since the errors, in this condition, are all underestimation of the correct numerosity compared to the Moderate condition, where the Delta value is closer to zero. Moreover, post-hoc analysis of Synchrony revealed the worst performance in the Asynchronous condition (Delta value: -1.98) compared to Synchronous (Delta value: -0.52) ($BF_{10} > 100$, posterior odds = $2.76 \times 10^{11}$) and to No sound (Delta value: -0.79) conditions ($BF_{10} > 100$, posterior odds = $1.54 \times 10^{9}$). There was no evidence for differences in the Synchronous and No Sound conditions ($BF_{10} = 0.65$, posterior odds = 0.38).

The post hoc analyses on the effect of Synchrony show that participants were more accurate when auditory and visual stimuli were synchronized. The asynchronous presentation of auditory and visual stimuli makes participants lose more counts, so it appears that the auditory stimulus acts as an exogenous signal interfering with the counting process but not operating as a trigger and activating the next count.

As regarding the interaction "Isochrony * Speed" (see Fig 3), the post-hoc analysis revealed decisive evidence for a better performance in the Isochronous–Moderate condition (Delta value: -0.31) as compared to the Isochronous—Fast condition (Delta value: -1.77) ($BF_{10} > 100$, posterior odds = 235038.78); and a better performance in the Non-isochronous—Moderate condition (Delta value: 0.15) as compared to the Non-isochronous—Fast condition (Delta value: -2.46) ($BF_{10} > 100$, posterior odds = $7.44 \times 10^{12}$). Such effects showed in the first place that participants most underestimated the count when the stimuli presentation was faster than when it was slower (moderate) both in the case of Isochronous and Non-isochronous conditions, although the interaction revealed that such effect was greater in the Non-isochronous than the Isochronous conditions (see Fig 3). Secondly, with these two comparisons we can see different patterns of results. Indeed, when comparing Delta data of Moderate (-0.31) and Fast

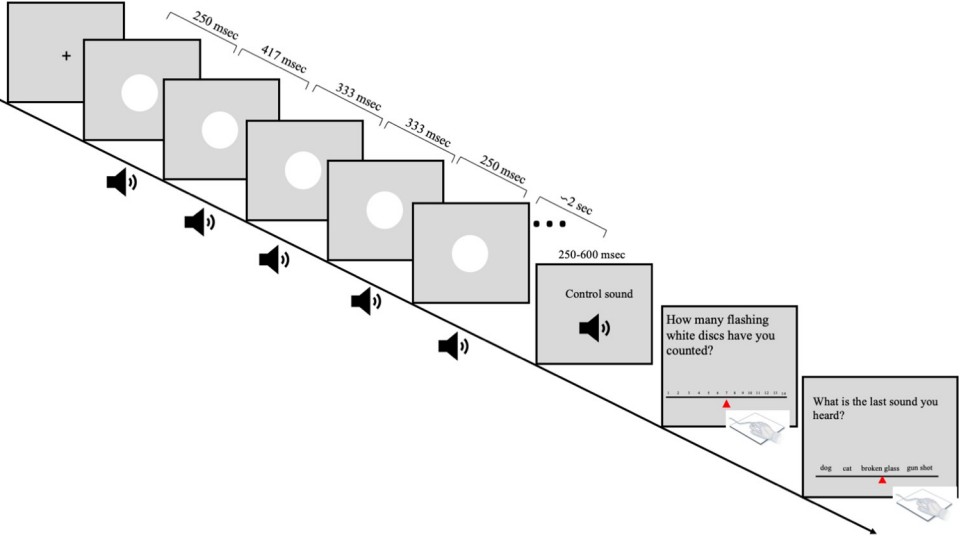

**Fig 3. Graphical representation of one trial in the fast, non-isochronous, synchronous condition.**

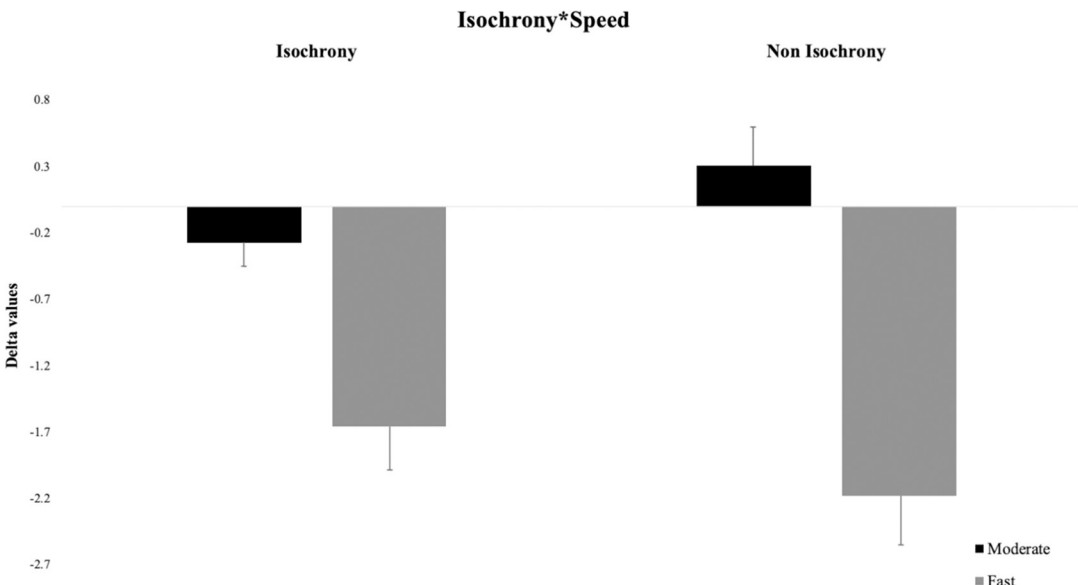

**Fig 4. Bar graph displaying Delta data as a function of isochrony and speed in Experiment 1.** The error bars represent the standard error.

(-1.77) Isochronous conditions, it emerges that in each case these are underestimation errors. On the other hand, when comparing Delta data of Moderate (0.15) and Fast (-2.46) Non-isochronous conditions it emerges that when the presentation is moderate, the errors move towards overestimation, on the contrary, when the presentation is fast there are larger underestimation errors. Moreover, the interaction "Isochrony*Speed" revealed an increase of underestimation of faster stimuli when their presentation was not isochronous (Delta value: -2.46) compared with when it was isochronous (Delta value: -1.77), and an opposite pattern of results (increase of overestimation) with slower stimuli when their presentation was not isochronous (Delta value: 0.15) compared with when it was isochronous (Delta value: -0.31). In other words, the aspect that changes in the Fast vs. the Moderate conditions is the direction of the results. Indeed, in the Moderate Isochronous condition the errors made by the participants go in the direction of an underestimation while in the Non-isochronous Moderate condition the errors move more towards an overestimation. In summary, participants make more underestimation errors when they count stimuli presented slowly with regular intervals, and, on the other hand, they make more overestimation errors when stimuli are presented slowly with irregular intervals (see Fig 4). Therefore, in general, it seems that speed always produces underestimation, but when instead non-isochrony is analyzed specifically, speed and isochronicity have a different way of interacting: When intervals of presentation are regular, participants lose counts, while when they are irregular, participants seem to count more items than those presented.

## Experiment 2

The results of Experiment 1 show that speed is a relevant factor in the counting process, leading participants to generate more undercount errors. Since there seems to be a linear effect of speed, we conducted a second experiment to further explore this hypothesis. In the Experiment 2 we added the "Slow" condition, in which there were 1.5 flashes per second on average.

## Method

**Participants.** Twenty-six participants (5 males, mean age = 22.07 years; SD = 3.5 years; range = 19–30) took part in the experiment for course credit. All the participants reported normal or corrected-to-normal vision and audition. The participants were naïve as to the purpose of the study and gave informed written consent. Authors had access to information (Name and Surname, university registration number, gender, hours of sleep in the night before doing the experiment, age) that could identify individual participants during and after data collection.

**Materials.** Thirty-six video or audio-video sequences were created for the experiment. All sequences had the same fps rate, background, visual stimuli (flashing white discs), and auditory stimuli than the ones used in Exp. 1. Eighteen sequences were the same "Moderate" sequences used in Exp. 1, while the remaining 18 were slower ("Slow"). The number of flashes in each sequence varied pseudo-randomly (average: 31, min: 26, max: 34). In half the sequences the flashes were isochronous (with regular time interval), while in the other half they were irregular. In this case, half of the sequences was "Moderate" (2 flashes per second on average) while the other half was "Slow" (1.5 flashes per second on average) and varied in length according to the total number of flashes (average length, in seconds, of both "Moderate" and "Slow" sequences: 15.3, min: 13.6, max: 17.4). The sequences varied also according to sound presence and type, as in Exp. 1: "No sound", "Synchronous", and "Asynchronous" sequences had the same features described in Exp. 1. Similarly to Exp. 1, stimuli created 12 different conditions: 1. White disc flashing isochronously with no sound, moderate (2/s); 2. White disc flashing isochronously with no sound, slow (1.5/s); 3. White disc flashing isochronously with synchronous "tick" sounds, moderate (2/s); 4. White disc flashing isochronously with synchronous "tick" sounds, slow (1.5/s); 5. White disc flashing isochronously with asynchronous "tick" sounds, moderate (2/s); 6. White disc flashing isochronously with asynchronous "tick" sounds, slow (1.5/s); 7. White disc flashing non-isochronously with no sound, moderate (2/s); 8. White disc flashing non-isochronously with no sound, slow (1.5/s); 9. White disc flashing non-isochronously with synchronous "tick" sounds, moderate (2/s); 10. White disc flashing non-isochronously with synchronous "tick" sounds, slow (1.5/s); 11. White disc flashing non-isochronously with asynchronous "tick" sounds, moderate (2/s); 12. White disc flashing non-isochronously with asynchronous "tick" sounds, slow (1.5/s).

Each condition had 3 instances, for a total of 36 trials. The 36 trials containing the 12 different conditions were presented in a random order. Every sequence was followed, after a random interval of approx. 2 sec, by 1 out of 4 "control" sounds: gunshot, dog barking, cat meowing, glass shattering. The control sounds had a length varying from 250 msec to 600 msec and their volume was set to perceptually match that of the "tick" sound.

**Procedure.** Same procedure used in Exp. 1.

**Apparatus.** Same apparatus used in Exp. 1.

## Results

As for Experiment 1, we include in the analyses only participants with accuracy at the control task above 90%, leading to the exclusion of three participants. Thus, analysis was performed on 23 participants (5 males; mean age = 22.3 years; SD = 3.7; range 19–30 years). The average accuracy of the control question on this subsample of subjects was 96% (SD = 0.02%) indicating that the participants clearly heard the sounds for the entire duration of the experiment.

The next analyses were performed on Delta values as in Experiment 1 (PC-CC). We ran a three-way repeated-measure Bayesian analyses of variance (ANOVAs; [126]) with Isochrony (2 levels), Speed (2 levels) and Synchrony (3 levels) as independent within-subject factors,

**Table 2. Analysis of the effects in the Bayesian ANOVA conducted on the Delta data in Experiment 2.**

| Effects | P(incl) | P(excl) | P(incl\|data) | P(excl\|data) | $BF_{10}$ |
|---|---|---|---|---|---|
| Isochrony | 0.263 | 0.263 | 0.192 | 0.737 | 0.260 |
| Synchrony | 0.263 | 0.263 | 0.652 | 0.04 | 173.21 |
| Speed | 0.263 | 0.263 | 0.636 | 0.001 | 428.31 |
| Isochrony ✳ Synchrony | 0.263 | 0.263 | 0.025 | 0.237 | 0.104 |
| Isochrony ✳ Speed | 0.263 | 0.263 | 0.051 | 0.212 | 0.239 |
| Synchrony ✳ Speed | 0.263 | 0.263 | 0.329 | 0.666 | 0.494 |
| Isochrony ✳ Synchrony ✳ Speed | 0.053 | 0.053 | $2.28 \times 10^{-4}$ | 0.001 | 0.166 |

P(incl) and P(excl) represent the priors of including and excluding the predictor before seeing the model. P(incl|data) and P(excl|data) represent the posteriors of including and excluding the predictor after seeing the model. $BF_{10}$ represents how much more likely the data are under the model including the predictor than under the model excluding the predictor.

using Delta data as a dependent variable. As in the previous experiment, correction for multiple comparisons in the post hoc test was performed using the approach discussed in Westfall [129]. All the statistical analyses were performed using JASP (Version 0.16.4, https://jasp-stats.org/).

Bayesian ANOVA revealed as the best model, the model containing the effect of Speed, and Synchrony ($BF_M$ = 17.7, see also Table 2, and Fig 5). The evidence in favor of the inclusion of the main effect of Speed and Synchrony was decisive (all $BF_{10} > 100$). On the contrary, there was no evidence for the main effect of Isochrony ($BF_{10}$ = 0.260) Moreover, there was no evidence in favor of the inclusion of all the other interactions ("Isochrony*Speed", $BF_{10}$ = 0.239; "Speed*Synchrony", $BF_{10}$ = 0.494; "Isochrony*Synchrony", $BF_{10}$ = 0.104; and the three-way interaction "Speed*Synchrony*Isochrony" $BF_{10}$ = 0.166).

Post-hoc analysis of the Speed effect revealed that participants were more accurate when the presentation of stimuli was slow (Delta value: 0.02) as compared to the Moderate condition (Delta value: -0.85), ($BF_{10} > 100$, posterior odds = 879.30). When the presentation of stimuli

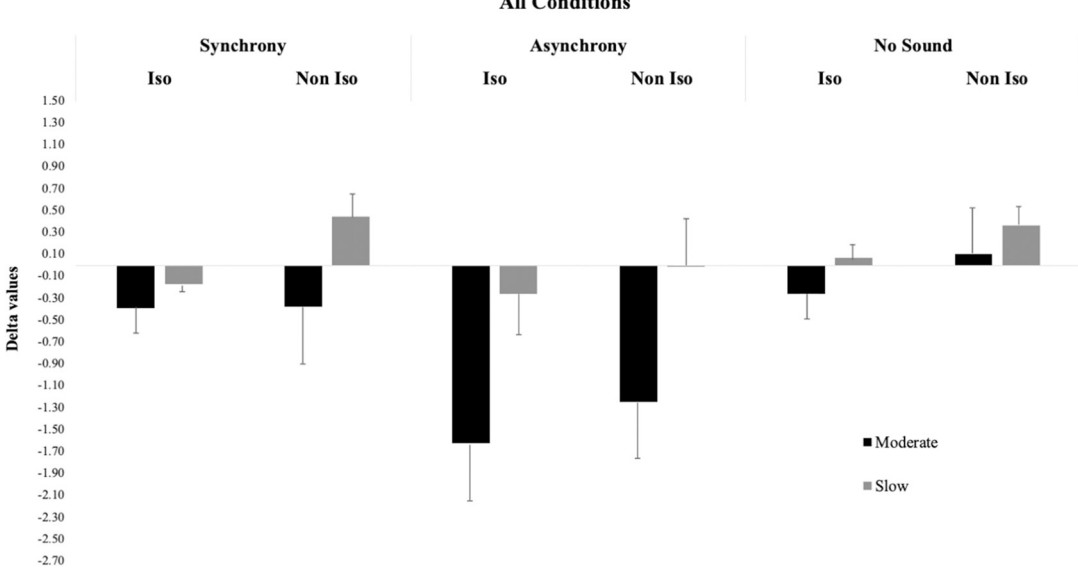

**Fig 5. Bar graph displaying Delta data of all conditions of Experiment 2.** The error bars represent the standard errors.

was moderate participants undercounted the stimuli, as compared to the Slow condition, where the Delta value is closer to zero. Moreover, post-hoc analysis of Synchrony revealed the worst performance in the Asynchronous condition (Delta value: -1.04) compared to Synchronous (Delta value: -0.25) ($BF_{10}$ = 9.02, posterior odds = 5.30) and to No sound (Delta value: 0.04) conditions ($BF_{10}$ > 100, posterior odds = 104.04). There was no evidence for differences in the Synchrony and No Sound conditions ($BF_{10}$ = 0.375, posterior odds = 0.220). The post hoc analyses on the effect of Synchrony show that participants were more accurate when auditory and visual stimuli were synchronized. Thus, the asynchronous presentation of auditory and visual stimuli makes participants lose more counts, confirming in this second Experiment as well that auditory stimuli interfered with the counting process without triggering additional counts.

## General discussion

In this study, across two experiments, we tested two different hypotheses regarding the possible detrimental effect of a concurrent auditory stimulus on a visual event counting task. Our results support the hypothesis that auditory stimuli break into the counting process, disrupting it, as indicated in literature (e.g. [53]) acting as an exogenous stimulation that attracts the attention of the participants. With our results it is therefore possible to confirm previous studies in which when temporal judgment is required auditory processing dominates over visual processing [35, 40–48]. Moreover, our results highlight a further case of crossmodal interference between visual and auditory domains in a specific task such as counting [26, 66–74].

The interesting result is precisely represented by the direction that the results have, and not by a generic inaccuracy of the participants. In fact, according to our hypothesis, the positive or negative direction can offer insight into the precise disruption that auditory stimuli have on visual ones [26, 66–74]. The disruption we found generally resulted in an underestimation of numerosity, namely, participants seem to lose their count instead of counting more items. Based on these results, therefore, the hypothesis in which it was postulated that auditory stimuli could act as a trigger by entering the counting process and making participants count more stimuli, as in the illusions described by Shams and colleagues [35], does not seem to be validated in this case.

Our results show that speed is a relevant factor in the counting process. We manipulated the speed of presentation of the stimuli by creating different conditions, in Experiment 1 we compared a Moderate condition and a Fast condition while in Experiment 2 we graduated the speed of presentation by comparing the Moderate condition (same as Experiment 1) with a new Slow condition. Results in general show that as speed increases the participants tend to deviate more from the correct count in a negative way, generating more errors in defect. This is also interesting in relation to the results of Experiment 2: the Slow condition compared with the Moderate condition, suggests a linear trend of the speed factor. This can be explained by the fact that when the presentation speed increases, the next stimulus appears (therefore must be counted) before the processing of the previous stimulus is completed (or the count updated or consolidated), resulting in the damage of the counting process itself (similarly to what happens in the phenomenon of the Psychological Refractory Period, see [131]). This outcome can be explained by a counting process based on an internal motor simulation (e.g., articulating the numbers), affected by speed. Thus participants, to complete the task, covertly articulate the current count thanks to the phonological loop and when the presentation of stimuli to be counted is fast the ideomotor articulation of the current count is cut short by the appearance of the next target to be counted (as happens with subvocal articulation of words in different languages, see [113, 114]). This mechanism seems to determine several lost counts and a general tendency to underestimation.

Results of both Experiments did not reveal a facilitatory effect of synchronous stimuli presentation compared to the absence of sound, thus not replicating a crossmodal facilitation effect found in other crossmodal interactions. Diederich and Colonius [132] provided evidence of faster response times (RTs) to monochromatic lights when they were accompanied by simple tones presented simultaneously, while Sinnett et al. [32] found that participants were faster to detect visual targets when they were presented together with sounds. Moreover, in literature it is possible to find enhancement of the salience of visual stimuli by temporally congruent auditory stimuli [79, 133, 134]. However, the supposed crossmodal enhancement of the synchronous condition compared to the soundless condition seems not applicable to a continuous, longer task. It is possible to hypothesize that our task demand is not difficult enough to highlight the presence of a facilitation effect.

In the extant literature (see for example [24]), there are other cases of this lack of facilitation effect within the debate on the specific modality or supramodal nature of the attentional system. For example, a study by Li et al. [135], in the field of implicit statistical learning (ISL) investigated whether multimodal input sources are processed separately or are integrated across modalities. Results reported that there did not highlight neither a facilitative learning effect nor an interference effect. These findings confirm theories that suggest the existence of multiple modality-specific mechanisms rather than a single central one. Another study reported that neural responses in the primary visual cortex can be reduced by auditory inputs that are synchronously presented with visual inputs [136]. Moreover, it seems that the facilitation effect of multisensory integration is strong when each unimodal signal is weak whereas this facilitation effect seems to decrease with increasing intensity of unimodal stimuli [28, 132]. It would therefore be interesting in future research to manipulate the intensity of crossmodal stimulations in a counting task, to verify for the occurrence of crossmodal facilitation.

The Synchrony main effect in both Experiment 1 and 2 also presents notable results. The asynchronous sequences seem to produce a worse performance than when the presentation is synchronous. Specifically, when visual and auditory stimuli are non-synchronized, participants tend to lose count. This also happens when the speed of presentation of the stimuli is very low, as indeed can be seen from the results of Experiment 2 in which crossmodal interference also emerges in a situation in which the task is facilitated, as it happens for example in the Slow condition, generating a better performance than in the Moderate condition. This result is in line with crossmodal studies where interference is generated by a lack of correspondence between auditory and visual stimuli. Demonstrations of this phenomenon include crossmodal Stroop effects from a stream of auditory distractors on responses to a concurrent stream of visual targets (e.g., [137–144]). Also, McDonald et al. [78], demonstrated that the occurrence of an irrelevant sound improved the detectability of a subsequent light when the target light appeared in the same location as the sound, while performance decreased when there was a spatial mismatch between the irrelevant sound and the target light (see also [124] for similar results with complex visual scenes). In our results, a crossmodal interference produced by the desynchronization between visual and auditory stimuli deteriorates the performance, increasing underestimation errors.

Underestimation becomes even more relevant when the stimuli are presented at a faster pace. Results of Experiment 1 show a different pattern of effects in the interaction between Isochrony and Speed. Namely, participants tend to lose more count in the Non-isochronous condition when stimuli are presented faster compared to all other interactions. This result confirms previous studies on fluency [99, 100] while offering evidence about the interaction with speed, which in our study proves to modulate the effects of irregularity. While in Experiment 1 participants make roughly the same number of errors when the presentation speed is moderate in all sequences, the direction of the error changes according to Isochrony: in the

isochronous condition there are more underestimation errors while in the Non-isochronous condition the errors tend more towards overestimation. This pattern of results can be interpreted considering the role of attention. When the interval of presentation of the stimuli is irregular the participants must pay more attention because they are unable to create expectations about the appearance of the following stimuli. For this reason, in the Non-isochronous condition presented slowly, the sequence calls for an increased focus to detect the unexpected appearance of the next stimulus. The percentage of error, even if in excess, is still very low. On the other hand, in the condition in which the presentation of the stimuli is isochronous and the speed moderate, the process can rely on the formation of expectations and entrainment, which could result in a percentage of error, albeit minimal, towards an underestimation. This hypothesis could be based on studies on the "Dynamic attending theory" [102–104]. Since the amount of errors in our Moderate condition is very low, we can say that when the presentation is also isochronous, therefore making the task even easier, the participants tend to be less accurate and to make errors by underestimation, while when the non-regularity of the intervals does not allow an attentional entrainment, participants lose this attentional facilitation and tend to get more confused in counting, thus counting by excess, as if they had illusorily perceived extra stimuli. The lack of significant results for the Isochronous condition in Experiment 2, together with the absence of a significant interaction with the Speed factor (differently from the effects recorded in Experiment 1), gives us some interesting insights. We interpret this result, that seems not to benefit from the regularity of stimuli presentation in the Slow condition, as actually a confirmation of the results of the first experiment. Indeed, as the speed increases, it seems that the participants benefit from the regularity of the presentation intervals, thanks to the creation of expectations over time that regularity provides [102–104]. In a condition where the presentation interval is slower, this metacognitive cue to support the task, as intended by Stevenson and Carlson's [100] definition, seems to be no longer necessary. In other words, in the second experiment in which the stimuli presentation speed is moderate or slow, we can assume that the participants no longer need to rely on the creation of temporal expectations (given by the regularity of the intervals) and therefore this variable no longer emerges as significant, either as a main effect or as a function of speed.

To sum up, there appears to be a linear relationship between errors and speed. More interestingly, as demonstrated in Experiment 1, it seems that errors that come from the speed factor could be mitigated by synchrony and enhanced by non-isochrony. We can interpret these results in line with the idea that in event counting, each step involves the integration of the perception of a visual stimulus and a representation of the next number in a counting sequence, associated with a motor plan involving covert articulation of the current count [100]. Thus, following Stevenson and Carlson's interpretation [100], tasks that require regular updating of events, like covert counting, come with a cost, namely, that running counts could be unstable. This instability might be partially compensated with metacognitive monitoring, which could occur implicitly: Cues like temporal fluency (regularity) may be used to create expectancies useful to stabilize the running count. It is possible to summarize by saying that there are some factors that make the task of counting complex, but as long as it is possible to rely on a metacognitive cue such as regularity, we can compensate for the detrimental effects of Speed and Asynchrony.

In general, the only trace of facilitation we observed in Experiment 1, although not crossmodal, is with moderate and isochronous sequences, both synchronized with sounds or without any sound at all, partially confirming previous literature [76–79]. The effect of Non-isochronous intervals emerged especially in the "Fast" conditions, in which the performances seem to be the worst. These results primarily show us that task-irrelevant sounds can exogenously capture our attention [124] with the power to also disrupt a longer process such as counting, resulting in missing some task-relevant stimuli.

Finally, with this study it is possible to add evidence to the phenomenon of competition for attentional resources between the visual and the auditory modalities (see [85–87]). In our study, auditory stimuli that are not synchronized with the simultaneous presentation of visual stimuli negatively affected the performance in a counting task. This could happen because the attentional resources are automatically captured by the auditory modality, making them temporarily unavailable to the visual processes and consequently affecting the counting process itself. Thus, this study add evidence in support of the supramodal attentional model [85–87].

In conclusion, it should be noted that some limitations are present. Firstly, due to COVID-19 prevention measures, all administrations were conducted online. While it is true that online experiments may not provide the same level of accuracy as laboratory administrations, with slightly more variability in all measurements, several studies suggest that modern online administration platforms are reliable in terms of accuracy and precision of presentation and response times [145, 146]. In addition, Bridges, Pitiot, MacAskill & Peirce [147] found that PsychoPy is generally one of the best platforms, achieving results very close to millisecond accuracy on different browser/operating system combinations. A second limitation may be associated with the fact that since the number of visual and auditory stimuli was the same in the bimodal condition, we cannot completely rule out the possibility that some participants mistakenly counted sounds instead of visual stimuli. Despite this, the fact that this misunderstanding of the task could have had a significant impact on results can be excluded, since the participants should have had to: 1) completely ignore instructions and training; 2) switch their task continuously (as the stimuli were randomized), counting visual stimuli in unimodal sequences and then sounds in bimodal stimuli. The probability that several participants conducted the experiment in this way is extremely thin. Finally, a third limitation might be associated with the length of the sequences. Although a change in the degree of vigilance is unlikely with sequences as short as the ones used in this study, it is nevertheless possible that different sequence lengths had a different impact on performance.

## Supporting information

**S1 Data.**
(XLSX)

**S2 Data.**
(XLSX)

## Author Contributions

**Conceptualization:** Claudia Del Gatto, Riccardo Brunetti.

**Data curation:** Claudia Del Gatto, Allegra Indraccolo, Tiziana Pedale.

**Formal analysis:** Allegra Indraccolo, Tiziana Pedale, Riccardo Brunetti.

**Investigation:** Claudia Del Gatto.

**Methodology:** Claudia Del Gatto, Allegra Indraccolo, Riccardo Brunetti.

**Project administration:** Allegra Indraccolo.

**Resources:** Riccardo Brunetti.

**Software:** Claudia Del Gatto.

**Supervision:** Claudia Del Gatto, Riccardo Brunetti.

**Validation:** Claudia Del Gatto.

**Visualization:** Claudia Del Gatto.

**Writing – original draft:** Claudia Del Gatto.

**Writing – review & editing:** Claudia Del Gatto, Allegra Indraccolo, Tiziana Pedale, Riccardo Brunetti.

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
