## [Decision Letter · Decision Letter 0]

4 Jun 2023

PONE-D-23-07755Crossmodal interference on counting performance: evidence for shared attentional resourcesPLOS ONE

Dear Dr. Del Gatto,

Thank you for submitting your manuscript to PLOS ONE. After careful consideration, we feel that it has merit but does not fully meet PLOS ONE’s publication criteria as it currently stands. Therefore, we invite you to submit a revised version of the manuscript that addresses the points raised during the review process.

We look forward to receiving your revised manuscript.

Kind regards,

Sergei Tugin, PhD

Academic Editor

PLOS ONE

Journal Requirements:

“We are grateful to KWS SAAT SE & Co. KGaA, Einbeck, Germany, for supporting this research by producing the genetic materials and conducting field experiments in Gondelsheim. INRAE, Paris, France, is also thanked for supporting this research by providing the French dent line “F353”.”

“FA - German Academic Exchange Service, Bonn, Germany (grant no. 91770158). https://www.daad.de/en/

FA - KWS SAAT SE & Co. KGaA, Einbeck, Germany. https://www.kws.com/corp/en/

FA, TM - University of Hohenheim (TG77). https://www.uni-hohenheim.de/en/english

“NO”

5. We note you have included a table to which you do not refer in the text of your manuscript. Please ensure that you refer to Table 4 in your text; if accepted, production will need this reference to link the reader to the Table.

Reviewers' comments:

Reviewer's Responses to Questions

**Comments to the Author**

1. Is the manuscript technically sound, and do the data support the conclusions?

Reviewer #1: Yes

Reviewer #2: Partly

Reviewer #3: Yes

2. Has the statistical analysis been performed appropriately and rigorously? 

Reviewer #1: Yes

Reviewer #2: Yes

Reviewer #3: Yes

3. Have the authors made all data underlying the findings in their manuscript fully available?

Reviewer #1: No

Reviewer #2: Yes

Reviewer #3: Yes

4. Is the manuscript presented in an intelligible fashion and written in standard English?

Reviewer #1: Yes

Reviewer #2: Yes

Reviewer #3: Yes

5. Review Comments to the Author

Reviewer #1: Overall, this is a great paper that reports on a novel finding regarding the effects of multisensory interference on counting performance. However, some parts may need to be revised or clarified.

Introduction:The literature review was generally very well conducted. Following are a few minor comments:

1. The connections between cross-modal interactions and event counting could be more explicitly pointed out in the introduction part. The sensory dominance, Mechanisms of cross-modal interference, and Attention in multi-sensory processing sections in the introduction part need to be linked to counting more.

2. Line 63-65: synchrony and temporal congruence seem in cross-modal integration seems to be most relevant to counting (Arrighi, R., Alais, D., & Burr, D. (2006). Perceptual synchrony of audiovisual streams for natural and artificial motion sequences. Journal of vision, 6(3), 6-6.)

3. Line 197-202: cross-modal entrainment in rhythmic multisensory stimuli could impact behavioral outcomes of perception subliminally (Lin, C., Yeh, M., & Shams, L. (2022). Subliminal audio-visual temporal congruency in music videos enhances perceptual pleasure. Neuroscience Letters, 779, 136623).

4. The authors could further clarify this relationship in the ‘Attention in multisensory processing’ and ‘Factors affecting selective counting’ sections: attention facilitate working memory as a buffer for both the encoding and retrieval of information necessary for performing cognitive tasks like counting (Awh, E., Vogel, E. K., & Oh, S. H. (2006). Interactions between attention and working memory. Neuroscience, 139(1), 201-208.).

Methods:

1. Why did you choose 3/second I the slow condition and 4/second in the fast condition? Any support for this choice?

2. The study was conducted online using PsychoPy. but it remains unclear how well its online version is able to maintain the temporal precision of the stimuli presentation, given that temporality of the stimuli is an important contributing factor to the results. Is there any studies on that?

3. Why did the authors choose Bayesian paradigm over the frequentist p-value? What are the advantages? This has been extensively reported, so the authors may want to include this rationale in the manuscript as well, given that the they had two competing hypotheses.

4. data was not included in the submission, so I was not able to validate the results.

Discussion:

1. A section about the limitation of the study is lacking. e.g. experiment conducted online

2. Also, why was the study conducted online? It seems that a stricter psychophysical protocol in lab may be helpful for reducing noises in the data.

Visualization:

1. A graph illustrating the experiment procedure would be appreciated.

2. The graphs lack y axes, not clear enough

Wording:

1. I suggest replacing ‘isochronous’ with ‘regular.’ Isochronous could also mean occurring at the same time, which may cause confusion.

Reviewer #2: I congratulate the authors, but I would like to suggest some important modifications before the renowned PLOS ONE journal can accept the manuscript.

In my opinion, examples should not be given in references and citations. Examples must be displayed in the body of the text. See lines 72, 77 and 82. If the authors exemplify the types of cognitive functions in the text, readers who do not understand neuropsychology and cognitive functions will be able to understand the article much better.

In the section METHODS, for material, as it is an innovative and online method, I suggest you post photos of the material used or some volunteers being evaluated.

Reviewer #3: Thank you for the opportunity to read and review “Crossmodal interference on counting performance: evidence for shared attentional resources”.

The manuscript is well-written, has a nice theoretical overview, and is generally an interesting study. The authors state that the data are available in the supplementary material, but I have not been able to access these, so my evaluation is based on the analysis and figures in the manuscript. While I enjoyed reading the manuscript, I have some concerns as to the design, analysis, and conclusions.

My major concern relates to the fast vs slow condition, the relatively large error bars in Fig 2, and the omission of data.

The study is an online study which has less control and more noise that a lab experiment. Naturally, online studies have become more of an everyday thing during covid, but this also provides challenges and a fair amount of additional uncertainty to the data. While the authors control for audition, there seem to be less control of visual stimuli and timing. This becomes an issue with the relatively large error bars (e.g., fig 2). Is it possible to gage how well visual stimulus and timing varied between participants?

Related the fast and slow condition (3/sec vs 4/sec) seem to be determined a bit at random, and as there seem to be a linear assumption in the manuscript it would be very nice to have at least one additional condition, either faster or slower (e.g., 2/sec) to have more certainty in the conclusion.

Finally, I think it is a very poor choice to exclude “outliers” (L340-341) it seems that 5% of the extremes were excluded, which is the natural extremes outside 2 SD from the mean (the criterion the authors used for the exclusion). In my opinion you need to have much better argument for excluding data and these should not be excluded from the analysis.

In addition I have some minor comments and suggestions;

L56-58 please add a reference.

L88-89 please link the sentence to the original authors presented in the line prior.

L99-100 it seems there may be a word missing in this line.

L181 Here the authors link the auditory dominance effect particularly to temporal tasks, perhaps this notion would be good to introduce when the auditory dominance effect is discussed previously.

L183 a “the” seem to be missing before authors.

L195-198 in discussing the effect of expectancy on attention you may consider some of the work by Kia Nobre and Jennifer Coull who have done extensive work in this area.

L205 Consider changing the wording from “see also, for crossmodal effects” to “for crossmodal effects see also” which I think is more correct wording (although English is not my native language).

L210 Consider changing “counting silently, covert counting,” to “counting silently (covert counting)”.

L221 Consider changing the wording from “no one has explicitly investigated this aspect to date in counting” to “to date no one has explicitly investigated this aspect in counting”.

L279 How were FPS controlled as the data was collected online – could this have varied between basic laptops and high-end gaming computers?

L279-282 How were participants instructed in order to ensure minimal variation around the estimated 9 degrees angle?

L285-286 What was the basis determining slow and fast sequences?

L287-288 Could the difference in total length of the sequences be affected by varying degrees of vigilance as the fast sequence is only 2/3’s of the slow?

L305-307 What is the motivation specific choice of the “control” sounds – particularly as some of these (e.g., a gunshot) presumably could be affecting arousal? Did you control whether the “control” sound affected the following performance?

L324 “we may deem this prospect as extremely unlikely”; did you consider to compare the proportion of trails where the reported count corresponded to the auditory sounds. This would perhaps provide a more objective argument, that this was not the case.

L353-354 I am not an expert in Bayesian statistics, however, from the philosophical point of view correcting for multiple comparisons while important for a frequentistic approach (as the alpha value is inflated by familywise error), but this does not seem relevant for a Bayesian approach where information is continuously updated.

6. PLOS authors have the option to publish the peer review history of their article (what does this mean?). If published, this will include your full peer review and any attached files.

Reviewer #1: No

Reviewer #2: **Yes: **Livia Stocco Sanches Valentin

Reviewer #3: No

---

## [Author Response · Author response to Decision Letter 0]

28 Jul 2023

Reviewer #1

Overall, this is a great paper that reports on a novel finding regarding the effects of multisensory interference on counting performance. However, some parts may need to be revised or clarified. Introduction: The literature review was generally very well conducted. Following are a few minor comments:

1. The connections between cross-modal interactions and event counting could be more explicitly pointed out in the introduction part. The sensory dominance, Mechanisms of cross-modal interference, and Attention in multi-sensory processing sections in the introduction part need to be linked to counting more. 

Thank you very much for this suggestion. We have added some references to the counting process in different sections of the introduction (Sensory dominance, lines 106-117; Crossmodal interference, lines 143-148; Attention, lines 188-192).

2. Line 63-65: synchrony and temporal congruence seem in cross-modal integration seems to be most relevant to counting (Arrighi, R., Alais, D., & Burr, D. (2006). Perceptual synchrony of audiovisual streams for natural and artificial motion sequences. Journal of vision, 6(3), 6-6.) 

We agree with you. We have added the reference (line 65) you kindly indicated.

3. Line 197-202: cross-modal entrainment in rhythmic multisensory stimuli could impact behavioral outcomes of perception subliminally (Lin, C., Yeh, M., & Shams, L. (2022). Subliminal audio-visual temporal congruency in music videos enhances perceptual pleasure. Neuroscience Letters, 779, 136623). 

Thank you for this indication and for this very recent reference. We have added it to the text (lines 225-230).

4. The authors could further clarify this relationship in the ‘Attention in multisensory processing’ and ‘Factors affecting selective counting’ sections: attention facilitate working memory as a buffer for both the encoding and retrieval of information necessary for performing cognitive tasks like counting (Awh, E., Vogel, E. K., & Oh, S. H. (2006). Interactions between attention and working memory. Neuroscience, 139(1), 201-208.). 

Thank you for this helpful hint, we have added it to the text (lines 163-168).

Methods:

1. Why did you choose 3/second I the slow condition and 4/second in the fast condition? Any support for this choice? 

Thank you for this observation. Actually, there was a clerical error: the slow condition is 2/second and the fast condition is 3/second. We corrected this in the whole paper. 

Moreover, the reason we chose these slow/fast conditions lies in the concept of "Spontaneous Motor tempo", namely the tempo of self-paced regular and repeated movements, and corresponds to the preferred and natural rhythm to execute isochronous motor actions (e.g. walking, or tapping a finger), which usually tends to cluster around 500–600 msec (Fraisse, 1982; Hammerschmidt, Fieler & Wöllner, 2021; Moelants, 2002). In our study, indeed, the slow condition with the isochronous stimuli corresponds to 2 events every second (1 event every 500msec). Moreover, based on the literature, a faster tempo goes around 330 msec, which implies 3 events every second. Of course, these speeds are relevant just for isochronous stimuli. Instead, with non-isochronous stimuli these speeds are on average (Madison & Paulin, 2010; Pereira et al., 2022). 

We have added this explanation in lines 250-254; and also lines 269-273.

2. The study was conducted online using PsychoPy. but it remains unclear how well its online version is able to maintain the temporal precision of the stimuli presentation, given that temporality of the stimuli is an important contributing factor to the results. Is there any studies on that? 

Thank you, we are aware of this issue. The study was conducted during the pandemic, specifically during the lockdown and this is the reason why it was administered online. However, we ensured the perfect temporal alignment of the stimuli by creating videos for every condition and repetition. All our materials are thus video files in which auditory and visual stimuli are therefore already synchronized. Furthermore, the experiments were administered online on PsychoPy (Pavlovia), that runs in most browsers and have been shown to have good capabilities with respect to the timing of experiments, such as reaction time measures in response to visual and auditory stimuli, to an acceptable level of precision. For a large-scale comparison for lab-based and web-based software across multiple operating systems and browsers, see Bridges et al. (2020) and Anwyl-Irvine, Dalmaijer et al. (2021). Moreover, there is a study that discusses the technical implication of online studies: Eerola, Armitage, Lavan & Knight (2021) Online Data Collection in Auditory Perception and Cognition Research: Recruitment, Testing, Data Quality and Ethical Considerations, Auditory Perception & Cognition, 4:3-4, 251-280, DOI: 10.1080/25742442.2021.2007718. Another study investigated how different web platforms and operating systems impact experiment presentation and data quality: Anwyl-Irvine, A., Dalmaijer, E.S., Hodges, N. et al. Realistic precision and accuracy of online experiment platforms, web browsers, and devices. Behav Res 53, 1407–1425 (2021). https://doi.org/10.3758/s13428-020-01501-5. The results of these studies suggest that modern online administration platforms are reliable in terms of accuracy and precision for presentation and response times.

3. Why did the authors choose Bayesian paradigm over the frequentist p-value? What are the advantages? This has been extensively reported, so the authors may want to include this rationale in the manuscript as well, given that they had two competing hypotheses.

Thank you for this comment. We chose the Bayesian approach over the frequentist p-value because it permits not only to identify whether an effect is present or not but also the strength of evidence in favor or against such an effect. Indeed, while classic frequentist p-value allows only to accept or reject hypotheses, Bayesian ANOVA, by means of model comparisons, allows identifying the best model describing the data, and allows estimating the strength of evidence in favor (and against) the inclusion of each factor and each interaction involved in the model (i.e., no evidence, anecdotal, substantial, strong, very strong, or decisive).

The Bayesian advantage over frequentist p-values has been clarified on lines 400-403.

4. Data was not included in the submission, so I was not able to validate the results. 

We are sorry, obviously something went wrong in the submission as we have added the table with the data but evidently in the wrong section. We have now included the table with all the data.

Discussion:

1. A section about the limitation of the study is lacking. e.g. experiment conducted online.

Thank you for highlighting this point, we have added a “limitations” section (lines 702-719).

2. Also, why was the study conducted online? It seems that a stricter psychophysical protocol in lab may be helpful for reducing noises in the data. 

Thanks for this note. As mentioned above, the study was conducted during the lockdown, we used Pavlovia which, as indicated by the studies mentioned above, is considered a reliable administration tool especially in relation to administration times and recording reaction times.

Visualization:

1. A graph illustrating the experiment procedure would be appreciated. 

Thank you, we added a graphical representation of trials (Figure 2).

2. The graphs lack y axes, not clear enough. 

Thanks, we have now fixed the graph by adding the label on the y-axis.

Wording:

1. I suggest replacing ‘isochronous’ with ‘regular.’ Isochronous could also mean occurring at the same time, which may cause confusion. 

Thank you. We have chosen this terminology because it is the standard in literature about rhythm and dynamic attending. Instead of replacing “isochronous” with “regular” we have specified multiple times (lines 212, 329, 500) that with the term isochronous we mean regular time intervals while for events that occur simultaneously or not, we have used the term synchronous and asynchronous.

Reviewer #2

I congratulate the authors, but I would like to suggest some important modifications before the renowned PLOS ONE journal can accept the manuscript.

In my opinion, examples should not be given in references and citations. Examples must be displayed in the body of the text. See lines 72, 77 and 82. If the authors exemplify the types of cognitive functions in the text, readers who do not understand neuropsychology and cognitive functions will be able to understand the article much better. 

Thanks for this note, we have clarified some points in the text as you suggested (lines 72-75; 106-117; 143-148; 163-168)

1. In the section METHODS, for material, as it is an innovative and online method, I suggest you post photos of the material used or some volunteers being evaluated. 

Thank you for your suggestion. We have now added an image that illustrates the task (Figure 2) and another image that illustrates examples of the sequences we used (Figure 1).

Reviewer #3

Thank you for the opportunity to read and review “Crossmodal interference on counting performance: evidence for shared attentional resources”.

The manuscript is well-written, has a nice theoretical overview, and is generally an interesting study. The authors state that the data are available in the supplementary material, but I have not been able to access these, so my evaluation is based on the analysis and figures in the manuscript. While I enjoyed reading the manuscript, I have some concerns as to the design, analysis, and conclusions.

1. My major concern relates to the fast vs slow condition, the relatively large error bars in Fig 2, and the omission of data.

The study is an online study which has less control and more noise that a lab experiment. Naturally, online studies have become more of an everyday thing during covid, but this also provides challenges and a fair amount of additional uncertainty to the data. 

Thank you, we have added, in lines 250-254 a more detailed explanation of why we chose these fast and moderate (previously called “slow”) conditions. Also, as you suggest, we ran a second Experiment including a slower condition (now called “Slow”) to further support the linear hypothesis that emerges from the results of Experiment 1.

Finally, we added a limitations section in which we highlight the potential implications of an online administration (lines 702-719).

2. While the authors control for audition, there seem to be less control of visual stimuli and timing. This becomes an issue with the relatively large error bars (e.g., fig 2). Is it possible to gage how well visual stimulus and timing varied between participants? 

Thanks for this observation. It is unfortunately unclear for us if you mean a “control” in the sense of something that can verify if the participants are indeed looking at the visual stimuli and listening to the auditory stimuli, or a “control” as a comparison measure lacking the experimental manipulation.

If you mean the former: Since the visual stimuli are precisely our target, in other words the visual stimuli had to be counted by the participants, we did not add a specific control measure in this sense. But the Delta calculation tells us that the participant counts deviated from the correct count by -0.86 assuming 0 as total equivalence between the correct count and the participant’s count. Surely, as you rightly noted, the error bars are wide and there is variability between the evaluations, especially among those considered more difficult, i.e. those in which the participants made more errors and the delta values differ more from 0. But having an average of deviation from the correct count of -0.86 (less than 4%), we feel confident that this can be an appropriate control measure.

If you mean “control” as a comparison measure without experimental manipulation, while for task irrelevant stimuli, i.e. sounds, it is possible to eliminate them altogether as a control (as we did), the same is not possible for visual stimuli. Since visual stimuli is task-relevant, the only “control” we can use is the simplest condition of all, therefore the moderate regular with no audio – that we can intend as a baseline measure. Nevertheless, to increase the control (in this latter sense) even more, we added a second experiment, as you suggested, in which we used a slower condition (now called “Slow”, while the slower condition in Exp 1 is now called “Moderate”). In this way we can see whether what we considered a baseline measure of visual stimuli is a good control measure or whether there was an even simpler baseline possible.

Regarding the timing of stimuli presentation, this was perfectly adjusted as the auditory and visual stimuli were aligned upstream from us: all the stimuli were encoded as video sequences. 

3. Related the fast and slow condition (3/sec vs 4/sec) seem to be determined a bit at random, and as there seem to be a linear assumption in the manuscript it would be very nice to have at least one additional condition, either faster or slower (e.g., 2/sec) to have more certainty in the conclusion. 

Thank you for this observation. Actually, there was a clerical error: the slow condition is 2/second and the fast condition is 3/second. We corrected this in the whole paper. 

Moreover, the reason we chose these slow/fast conditions lies in the concept of "Spontaneous Motor tempo", namely the tempo of self-paced regular and repeated movements, and corresponds to the preferred and natural rhythm to execute isochronous motor actions (e.g. walking, or tapping a finger), which usually tends to cluster around 500–600 msec (Fraisse, 1982; Hammerschmidt, Fieler & Wöllner, 2021; Moelants, 2002). In our study, indeed, the moderate condition with the isochronous stimuli corresponds to 2 events every second (1 event every 500msec). Moreover, based on the literature, a faster tempo goes around 330 msec, which implies 3 events every second. Of course, these speeds are relevant just for isochronous stimuli. Instead, with non-isochronous stimuli these speeds are on average (Madison & Paulin, 2010; Pereira et al., 2022). 

We have added this explanation in lines 250-254 and 269-273.

Finally, we would like to thank you for your suggestion about a second experiment to explore more in depth the linear assumption. We added Experiment 2 with the condition “Slow”, in which we have on average 3 events every 2 seconds. According to Madison & Paulin, (2010), indeed, slow condition implies events up to 95 bpm.

4. Finally, I think it is a very poor choice to exclude “outliers” (L340-341) it seems that 5% of the extremes were excluded, which is the natural extremes outside 2 SD from the mean (the criterion the authors used for the exclusion). In my opinion you need to have much better argument for excluding data and these should not be excluded from the analysis. 

Thank you for this observation. We apologize if in the original version of the manuscript the criterion for subject exclusion was not clarified. In the original version of the manuscript, we excluded subjects according to two criteria: 1) accuracy at the control task (identification of control sounds) below 90%; and 2) a Delta score more than 2 SD from the mean. In the current version of the manuscript, we have now included all the subjects, even with a Delta score outside 2 SD from the mean. Notably, the inclusion of these subjects confirmed our previous results (please see lines 387-392 and lines 530-534). However, we did not include subjects with accuracy at the control task below 90%. This led to the exclusion of 2 participants in Experiment 1, and 3 participants in Experiment 2. We maintained this criterion because we aimed to include in the analyses only those participants that clearly heard the sounds for the entire duration of the experiment. Importantly, the maintenance of this criterion is even more important with our experiment since it was conducted online. 

In addition I have some minor comments and suggestions;

1. L56-58 please add a reference. 

Thank you, we added the appropriate reference (lines 56-58).

2. L88-89 please link the sentence to the original authors presented in the line prior. 

Thanks, we changed the sentence as suggested (line 88).

3. L99-100 it seems there may be a word missing in this line. 

Thanks, we corrected the sentence (line 101). 

4. L181 Here the authors link the auditory dominance effect particularly to temporal tasks, perhaps this notion would be good to introduce when the auditory dominance effect is discussed previously. 

Thanks, we modified the text moving this reference in the Sensory dominance section (lines 114-117).

5. L183 a “the” seem to be missing before authors. 

Thanks, we modified the sentence (line 115).

6. L195-198 in discussing the effect of expectancy on attention you may consider some of the work by Kia Nobre and Jennifer Coull who have done extensive work in this area. 

Thank you for this helpful suggestion, we added a reference (line 219). 

7. L205 Consider changing the wording from “see also, for crossmodal effects” to “for crossmodal effects see also” which I think is more correct wording (although English is not my native language). 

Thank you for this note, we changed the sentence accordingly (line 234).

8. L210 Consider changing “counting silently, covert counting,” to “counting silently (covert counting)”. 

Thank you, we changed the sentence accordingly (line 239).

L221 Consider changing the wording from “no one has explicitly investigated this aspect to date in counting” to “to date no one has explicitly investigated this aspect in counting”. 

Thank you, we changed the sentence accordingly (lines 254-255).

9. L279 How were FPS controlled as the data was collected online – could this have varied between basic laptops and high-end gaming computers? 

The Fps we chose was strategically low, this precisely to avoid possible lag and frame dropping. The resulting video files used in the experiments have a frame rate of 12 per second, which is considerably lower than the usual 24, 29 or 30 usually used in videos available on the internet.

10. L279-282 How were participants instructed in order to ensure minimal variation around the estimated 9 degrees angle? 

Thank you for this note. We confirm that the instructions required participants to sit at approximately 60 cm from the screen. We agree that it was certainly not possible to control this variable accurately. However, the size of the visual stimulus was kept constant during the experiment, and it was irrelevant to the task. Even if we assume that the participant varied their distance from the screen to perform the task, we can assume this variation was random, thus not affecting the results systematically. We have added this specification in the materials section (lines 323-328).

11. L285-286 What was the basis determining slow and fast sequences? 

Thank you for making us explain this point better. The reason we chose these slow/fast conditions lies in the concept of "Spontaneous Motor tempo", namely the tempo of self-paced regular and repeated movements, and corresponds to the preferred and natural rhythm to execute isochronous motor actions (e.g. walking, or tapping a finger), which usually tends to cluster around 500–600 msec (Fraisse, 1982; Hammerschmidt, Fieler & Wöllner, 2021; Moelants, 2002). In our study, indeed, the slow condition (now called “Moderate”) with the isochronous stimuli corresponds to 2 events every second (1 event every 500msec). Moreover, based on the literature, a faster tempo goes around 330 msec, which implies 3 events every second. Of course, these speeds are relevant just for isochronous stimuli. Instead, with non-isochronous stimuli these speeds are on average (Madison & Paulin, 2010; Pereira et al., 2022).

See lines 269-273.

12. L287-288 Could the difference in total length of the sequences be affected by varying degrees of vigilance as the fast sequence is only 2/3’s of the slow? 

Thank you. We have added this point in the limitations section (lines 702-719). While it is true that the length of sequences may have had a different impact on participants' performance, we believe that this cannot be associated with vigilance, since usually changes in vigilance levels emerge within much longer time frames.

13. L305-307 What is the motivation specific choice of the “control” sounds – particularly as some of these (e.g., a gunshot) presumably could be affecting arousal? Did you control whether the “control” sound affected the following performance? 

Thank you for raising this point. The volume of the control sounds was adjusted according to the volume of the auditory stimuli, specifically it was adjusted to a lower volume so as not to "alert the participants". In other words, the control sound was always at a lower volume in order to make it audible but to avoid increasing arousal levels. We have now included this specification (lines 355-358).

14. L324 “we may deem this prospect as extremely unlikely”; did you consider to compare the proportion of trails where the reported count corresponded to the auditory sounds. This would perhaps provide a more objective argument, that this was not the case.

The number of auditory stimuli is always the same as the visual ones. They were kept identical as a control measure. Unfortunately, this comparison cannot be performed since the number of stimuli is always the same. Thank you for highlighting this issue, we have added it in the limitations section (lines 702-719).

15. L353-354 I am not an expert in Bayesian statistics, however, from the philosophical point of view correcting for multiple comparisons while important for a frequentistic approach (as the alpha value is inflated by familywise error), but this does not seem relevant for a Bayesian approach where information is continuously updated.

We thank the reviewer for this comment. Actually, as with frequentist inference, also Bayesian post hoc tests are subject to the problem of multiple comparisons (please, see, van den Bergh et al., 2020; for an extensive description of the problem). Specifically, it is the estimation of the posterior odds that suffers from multiple comparison errors. Here, as it is implemented in JASP, we used the approach discussed in Westfall (1997) to correct for multiple comparison problems. Such an approach consists of the following steps: firstly Bayesian t-tests are computed for all pairwise comparisons, thus providing unadjusted Bayes factors; as a second step, the prior model odds are adjusted by fixing to 0.5 the prior probability that the null hypothesis holds across all comparisons; finally, the adjusted prior odds and the Bayes factor are used to calculate the adjusted posterior odds.

This additional information on the data analysis, and the references on the multiple comparison problems with post-hoc Bayesian analyses, have been added to the revised version of the manuscript (please, see lines 400-403 and lines 408-413).

---

## [Decision Letter · Decision Letter 1]

24 Aug 2023

PONE-D-23-07755R1Crossmodal interference on counting performance: evidence for shared attentional resourcesPLOS ONE

Dear Dr. Del Gato,

Thank you for submitting your manuscript to PLOS ONE. After careful consideration, we feel that it has merit but does not fully meet PLOS ONE’s publication criteria as it currently stands. Therefore, we invite you to submit a revised version of the manuscript that addresses the points raised during the review process.

We look forward to receiving your revised manuscript.

Kind regards,

Sergei Tugin, PhD

Academic Editor

PLOS ONE

Journal Requirements:

Reviewers' comments:

Reviewer's Responses to Questions

**Comments to the Author**

1. If the authors have adequately addressed your comments raised in a previous round of review and you feel that this manuscript is now acceptable for publication, you may indicate that here to bypass the “Comments to the Author” section, enter your conflict of interest statement in the “Confidential to Editor” section, and submit your "Accept" recommendation.

Reviewer #1: All comments have been addressed

Reviewer #2: All comments have been addressed

2. Is the manuscript technically sound, and do the data support the conclusions?

Reviewer #1: Yes

Reviewer #2: Yes

3. Has the statistical analysis been performed appropriately and rigorously? 

Reviewer #1: Yes

Reviewer #2: Yes

4. Have the authors made all data underlying the findings in their manuscript fully available?

Reviewer #1: Yes

Reviewer #2: Yes

5. Is the manuscript presented in an intelligible fashion and written in standard English?

Reviewer #1: Yes

Reviewer #2: Yes

6. Review Comments to the Author

Reviewer #1: I would like to thank the authors for their edits. All comments raised in the previous round of review have been addressed. The paper is now ready to be accepted.

Reviewer #2: Congratulations again to the authors. This corrected version of the manuscript is worthy of publication. I kindly ask you to adjust just two more minor details. In the METHOD section, in lines 310 and 491, start the sentences with the numerals in full. "Seventy-nine" participants... and "Twenty-six" participants...

All other requests were granted.

7. PLOS authors have the option to publish the peer review history of their article (what does this mean?). If published, this will include your full peer review and any attached files.

Reviewer #1: No

Reviewer #2: **Yes: **Livia Stocco Sanches Valentin

---

## [Author Response · Author response to Decision Letter 1]

30 Aug 2023

Reviewer #1: I would like to thank the authors for their edits. All comments raised in the previous round of review have been addressed. The paper is now ready to be accepted.

We thank you very much for your review work which has allowed us to greatly improve our study.

Reviewer #2: Congratulations again to the authors. This corrected version of the manuscript is worthy of publication. I kindly ask you to adjust just two more minor details. In the METHOD section, in lines 310 and 491, start the sentences with the numerals in full. "Seventy-nine" participants... and "Twenty-six" participants...

All other requests were granted.

We thank you very much for your review work which has allowed us to greatly improve our study. We changed the sentences in lines 310 and 491.

---

## [Editor Report · Decision Letter 2]

25 Oct 2023

Crossmodal interference on counting performance: evidence for shared attentional resources

PONE-D-23-07755R2

Dear Dr. Del Gato,

We’re pleased to inform you that your manuscript has been judged scientifically suitable for publication and will be formally accepted for publication once it meets all outstanding technical requirements.

Kind regards,

Sergei Tugin, PhD

Academic Editor

PLOS ONE

---

## [Editor Report · Acceptance letter]

26 Oct 2023

PONE-D-23-07755R2 

Crossmodal interference on counting performance: Evidence for shared attentional resources 

Dear Dr. Del Gatto:

I'm pleased to inform you that your manuscript has been deemed suitable for publication in PLOS ONE. Congratulations! Your manuscript is now with our production department. 

Kind regards, 

on behalf of

Dr Sergei Tugin 

Academic Editor

PLOS ONE